# Factors Associated with Health-Related Quality of Life among Government Employees in Putrajaya, Malaysia

**DOI:** 10.3390/ijerph18052626

**Published:** 2021-03-05

**Authors:** Muhamad Hasrol Mohd Ashri, Hazizi Abu Saad, Siti Nur’Asyura Adznam

**Affiliations:** 1Department of Nutrition, Faculty of Medicine and Health Sciences, Universiti Putra Malaysia, Serdang, Selangor 43400, Malaysia; muhamadhasrolmohdashri@gmail.com; 2Department of Dietetics, Faculty of Medicine and Health Sciences, Universiti Putra Malaysia, Serdang, Selangor 43400, Malaysia; asyura@upm.edu.my; 3Malaysian Research Institute of Ageing, (MyAgeing) Universiti Putra Malaysia, Serdang, Selangor 43400, Malaysia

**Keywords:** health-related quality of life, factors associated, government employees, Malaysia

## Abstract

The current rapid growth of the economy has necessitated an assessment of health-related quality of life (HRQOL) and its associated factors among employees. Unfortunately, there are still limited data available in this area among the Malaysian working population in government sectors. The aim of this study was to evaluate the factors associated with HRQOL among government employees in Putrajaya, Malaysia. This cross-sectional study recruited 460 eligible government employees who worked in the area of Putrajaya through simple random sampling. The self-administered questionnaire was distributed to these participants to collect information on the SF-36 profile of scores, sociodemographic factors, lifestyle factors, and medical history. The results of this study signify that most of the participants were identified as having good HRQOL with the mean score of overall HRQOL was 72.42 ± 14.99. Multivariate analysis showed that being younger, receiving a better monthly personal income, a smaller household number, performing more physical activity, not having any chronic disease, and not using any long-term medication were significantly positively associated with overall HRQOL. The participants who did not have a family history of chronic disease were reported to be significantly associated with better mental component summary (MCS). Further, males were significantly positively associated with bodily pain (BP) and general health (GH) only, whereas better occupational status was limited to social functioning (SF). In conclusion, the results of this study provide motivation for future research and initiatives for improving the physical, emotional, and social well-being of government employees.

## 1. Introduction

Health is defined as not only the state of an individual being free from disease or weakness but also includes the state of an individual’s physical, mental, and social well-being that can adequately cope with the demand of daily life and is compatible with a fulfilling life [1]. Health-related quality of life (HRQOL) is a person’s satisfaction towards their function and well-being in regard to the multi-dimensional aspects of life, which are affected by health as described [2]. Functioning here refers to the capacity of a person to complete a predetermined task or activity, whereas well-being alludes to a person’s inner feeling [2]. Notably, an accurate determination of HRQOL can yield comprehensive data on overall health state, which can alert public health officials to the burden of physical and emotional health as well as preventable disease, disablement, and injury in the workplace [3,4].

The assessment of HRQOL is not only seen as a relevant outcome measure for patient populations but is also an accepted metric for healthy groups of people, including employees who work in various workplaces and environments [5]. In this era of globalisation, a system of administration and technologies at work were evolved in line with rapid economic development, directly making employees valuable resources to an organisation [6]. Furthermore, the current workforce is no longer male-dominated; females also contribute to work-related fields in this modern world which has created new work environments and work demands [6]. Work is known as a social activity that can positively or inversely influence HRQOL [7]. Additionally, even though employees usually tend to be healthy, assessing their quality of life in terms of the health aspect is worthwhile because poor health status is associated with the loss of talented workers, sick leaves, reduced work efficiency, and early retirement as a result of medical issues [5].

In the daily life of most working adults, there is a substantial effect of atmosphere at the workplace on their physical and mental prosperity since so much time and energy is spent at the workplace [8]. Various conditions or activities of work, such as strain due to physical tasks, occupational stress, or exposure to hazards or accidents, contribute to the low level of HRQOL or work-related health issues (for example, disorders of the musculoskeletal system, body weight status, and psychological difficulties) [9,10]. The outcome of HRQOL among employees varies according to sociodemographic factors, such as sex, age, marital status, and socio-economic status [9,11,12]. On the other hand, a job position, as determined by the differences in job scope and demands at work, significantly impacts physical and emotional well-being [13,14]. The endpoint of quality of life in regards to health is also influenced by lifestyle factors of the worker, such as level of physical activity and smoking status [15,16]. Moreover, it has been accepted that there is an increasing prevalence of chronic diseases among the community, and hence burden due to medical history has a significant impact on numerous aspects of a worker’s life, such as their HRQOL [12,16].

Nowadays, consistent with most nations, Malaysia has focused on strengthening the economy and professionalism and competition in the workplace, hence pushing employees to produce more quality outcome [17]. Thus, ensuring optimum HRQOL among employees is essential not only for sustainable development of the economy, but also for political harmony and social prosperity. Conjointly, securing the well-being of physical, psychological, and social health of employees assists in stabilising an organisation. However, the majority of the published studies in Malaysia have considered only factors associated with HRQOL among the population with clinical presentation and the elderly; there is still a lack of data available concentrating on the HRQOL of the working population [18,19,20,21,22,23]. Therefore, the present study aimed to determine HRQOL and its associated factors—namely, sociodemographic factors, lifestyle factors, and medical history—among government employees in Putrajaya, Malaysia. The participants in this study worked in ministries that were mainly involved in administrative work, working eight hours per day and only the day shift.

An exploration on HRQOL can give employers or organisations insights to understand their workers’ health status. Thus, further action can be implemented to increase quality of work and job satisfaction and ensure the HRQOL of good workers throughout their tenure. Moreover, the findings of this study can also be used as a reference for public health policymakers, intervention programs, and future research to empower well-being of the working population and help achieve the nation’s health objectives. On the basis of the above explanation, we developed the research questions (RQs) of this study as stated below:

RQ1: What are the differences of HRQOL amongst government employees according to their sociodemographic factors, physical activity level, smoking status, and medical history?

RQ2: How do sociodemographic factors, physical activity level, smoking status, and medical history influence the HRQOL of government employees?

The structure of this paper is divided as follows. Materials and Methods: describes the methodologies of this study. Results: shows the presentation and explained the results and analyses. Discussion: comprises the discussion, and Limitations and Future Research: expresses the study limitations and future research. Lastly, Conclusions: provides the conclusion of this study.

## 2. Materials and Methods

This was a cross-sectional study conducted between August and November 2019, which applied a simple random sampling method to recruit government employees from six different ministries located in Putrajaya, Malaysia, as the study participants. The city of Putrajaya is a federal administrative capital and has also been recognised as a home for government employees and their families [24]. In this study, 1 ministry was randomly selected from each 6 different geographical areas in Putrajaya. A total of 540 government employees from a list of 3403 government employees working in those 6 ministries were randomly selected and invited to participate in the present study. All the invited government employees were screened first before being considered eligible study participants. Those who were under 18 years old, were non-permanent staff, were working less than 8 h/d, or had conceived were excluded. Ultimately, 460 of government employees were eligible as participants, signed the consent form, and completed the questionnaire. Thus, the overall response rate of this study was 85.2%. Prior to starting this study, ethical approval was obtained from the Ethics Committee for Research Involving Human Subjects, Universiti Putra Malaysia, reference no. UPM/TNCPI/RMC/1.4.18.2 (JKEUPM). Further, permission from the selected ministries and signed consent that confirmed involvement in this study from participants were obtained. The participants were also given an information sheet that explained the overview, benefits, and possible risks of the study. They were also instructed that the data would remain confidential.

A self-administered hard copy of a questionnaire, consisting of sections named sociodemographic, HRQOL, physical activity, smoking status, and medical history, was distributed to participants. Each questionnaire was checked and collected by researcher immediately after the participants finished answering in order to prevent any questions from remaining unanswered and to address missing items. The time for the participants to complete the questionnaire was around 20 min. In this study, selected sociodemographic factors, such as sex, age, marital status, ethnicity, educational level, occupational status, income per month, and household number were reported by the participants. Monthly household income was classified on the basis of the Department of Statistics Malaysia (DOSM) which is divided into lower, middle, and upper earnest known as the bottom 40% (B40), the middle 40% (M40), and the top 20% (T20), respectively [25].

The 36-item Short Form Health Survey (SF-36) is a widely used and well-recognised instrument to measure HRQOL. This instrument has been tested among many population groups of either healthy or unhealthy backgrounds in different countries for validity and reliability and has been translated into various languages; most of the results have been considered satisfactory. The present study determined the participant’s HRQOL by using the Malay version of the SF-36, validated by Sararaks et al. (2005) [26]. The SF-36 is composed of 36 items integrated into multi-item scales assessing 8 domains of health and a single item of health change (HC); hence, as a whole, it yielded, as a summary, a measure of overall HRQOL. The 8 domains of health were physical functioning (PF), the role of physical-related health (RP), bodily pain (BP), general health (GH), vitality (VT), social functioning (SF), the role of emotion-related health (RE), and mental health (MH). Among all the domains of health, the 4 domains of PF (10 items), RP (4 items), BP (2 items), and GH (5 items) represented the physical component summary (PCS); the other 4 domains—VT (4 items), SF (2 items), RE (3 items), and MH (5 items)—represented the mental component summary (MCS). The 36 items in this instrument used Likert scales and yes/no choices to evaluate happiness from the health-related function and well-being. Each of the items were coded, computed, and transformed according to its components, and a score was generated between 0 and 100, with a higher score indicating a more favourable health state.

Physical activity was measured by using the Malay version of the Global Physical Activity Questionnaire (GPAQ), validated by Soo et al. (2015) [27]. This instrument has a total of 16 questions and collects information on physical activity participation in 3 domains, namely, occupation-related physical activity, transport-related physical activity, and recreation-related physical activity, as well as sedentary behaviour. Metabolic equivalents (MET) were used in expressing the intensity of physical activity. The MET-minutes/week of physical activity in the domains and total physical activity were calculated on the basis of the equations as shown below:Vigorous activity: 8.0 MET × days/week × minutes/day = MET-minutes/week
Moderate activity: 4.0 MET × days/week × minutes/day = MET-minutes/week
Cycling or walking: 4.0 MET × days/week × minutes/day = MET-minutes/week
Total physical activity = sum of MET-minutes/week for each domain

The physical activity level of participants was classified into 3 categories:

High: At least 3 days of vigorous activity/week AND achieved at least 1500 MET-minutes/week of total physical activity OR at least 7 days of any combination of physical activity in the domains AND achieved at least 3000 MET-minutes/week of total physical activity.

Moderate: At least 3 days of vigorous activity/week AND included a total of at least 60 min/week of total activity OR at least 5 days of moderate activity and cycling or walking AND partook in these activities at least 150 min/week OR at least 5 days of any combination of physical activity in the domains AND achieved at least 600 MET-minutes/week of total physical activity.

Low: A value which did not fulfil the above-mentioned criteria was documented under this category.

Furthermore, smoking status was measured by using an adapted questionnaire from the National Health and Morbidity Survey (NHMS) in 2015 [28]. Through this instrument, current daily smokers, current occasional smokers, former daily smokers, former occasional smokers, those that have never smoked, and the number of cigarettes smoked by current smokers per day were identified. Participants were classified as a current smoker if they currently smoked one or more tobacco products per day or currently smoked tobacco products but not exactly every day. Participants were classified as a non-smoker if they currently do not smoke tobacco products, including former smokers and those who never smoked. In their medical history, participants were required to report whether they had been diagnosed with chronic diseases, were on long-term medication over the last 12 months, or had a family history of chronic diseases. Moreover, the participants who had been diagnosed with chronic diseases and/or had a family history of chronic diseases were asked which types.

The present study’s data were analysed using IBM SPSS Statistics for Windows, Version 25.0 (IBM, New York, NY, United States). Descriptive statistics for categorical data were described as frequency and percentage, meanwhile continuous data was described as mean and standard deviation. The differences in the HRQOL mean score between groups were determined by independent *t*-tests and one-way analysis of variance (ANOVA) tests. Multivariate linear regression was carried out to assess significant associations between various factors with HRQOL. All independent variables with less than a 0.25 significance level of association with the dependent variables in the bivariate analysis were included in the linear regression model. In this multivariate linear regression, independent variables, namely, age, income, household number, and physical activity were analysed as continuous variables; meanwhile, the others were analysed as categorical variables. In this study, associations of variables were statistically significant if the *p*-value was less than 0.05.

## 3. Results

Table 1 shows the characteristics of the participants, such as their socio-demographic, lifestyle, and medical history. In total, 460 participants were analysed in this study, 58.0% were female and 42.0% were male. The mean age of the participants was 36.32 ± 8.79 years, and more than half were 31–45 years (51.3%). Additionally, 65.0% of participants were married, Malay (86.3%), finished education at a tertiary level (72.2%), and worked as a support worker (55.4%). The mean monthly salary of the participants was MYR 3394.89 ± 1486.61 per month, with the largest fraction reporting an income of MYR 3000–4999 per month (41.1%). The mean household income obtained every month by the participants was MYR 5757.87 ± 3191.62, of which the majority of them were referred to as M40. This means they earned MYR 4850–10,959 of monthly household income (51.5%). The mean size of the household of the participants was 3.84 ± 2.05 persons, with the majority reporting less than five members in their house (54.3%). For lifestyle factors, the participants spent a mean total of 2751.59 ± 3188.10 MET-minutes/week on physical activity, for which most could be classified as a moderate physical activity level (48.7%). The results also showed that the majority of the participants were non-smokers (85.2%) compared to current smokers (14.8%). Among current smokers, the mean smoke intake was 10.68 ± 5.95 cigarettes per day. In the medical history, there was a high proportion of participants who had not been diagnosed with chronic diseases (82.8%), did not use medication long term (83.7%), and possessed a family history of chronic diseases (60.7%).

The SF-36 scores by the participants are shown in Table 2, and the mean score of overall HRQOL was 72.42 ± 14.99. The mean score of PCS and MCS were 73.36 ± 17.28 and 71.79 ± 14.13, respectively. For the domains of health in PCS, the highest mean score was RP (80.82 ± 27.37), whereas the lowest mean score was GH (59.79 ± 14.91). For the MCS’s domains of health, RE had a greater mean score (86.96 ± 24.18), and the VT obtained a lower mean score (64.92 ± 17.00).

Table 2 also presents the differences in the SF-36 profile of scores by sociodemographic characteristics of the participants. When considering sex, a significant difference was only seen in GH—males had a higher mean score than females. Further, participants who were in a younger age group, either single, divorced, or widowed, and possessed a tertiary educational level acquired a significantly greater mean score of overall HRQOL, PCS, and MCS than the other groups within its variables. Next, as shown in Table 3 regarding socio-economic status of the participants, those who worked as professionals had significantly higher mean score of overall HRQOL, PCS, and MCS compared to those who worked as support staff. Despite overall HRQOL and PCS, there was a significant difference in MCS’s mean score in relation to monthly personal income; participants who made less than MYR 3000 per month had a higher mean score than the other groups. In addition, the participants who were known as B40 for monthly household income and had less than five persons per household achieved a significantly larger mean score of overall HRQOL, PCS, and MCS.

Furthermore, Table 4 presents the mean score of SF-36 profile by lifestyle factors and medical history of the participants. In regard to lifestyle factors, this study found that there was significant higher mean score of overall HRQOL, PCS, and MCS among those who performed a high level of physical activity over those in the low and moderate groups. Moreover, as compared to non-smokers, current smokers reported a significantly greater mean score of overall HRQOL and PCS. For medical history, the mean score of overall HRQOL, PCS, and MCS were seen as higher among participants who did not have any chronic diseases or long-term medication use, and the difference was significant between groups. The participants who stated that they did not have a family history of chronic diseases were more likely to experience a significantly better mean score of overall HRQOL and MCS.

The multivariate linear regression was further performed to evaluate associations of various factors with HRQOL of the participants, as shown in Table 5. The participants who were younger in age and earned a better monthly personal income, had not been diagnosed with any chronic disease, and did not take any long-term medication were significantly positively associated with overall HRQOL, PCS, and MCS. Additionally, having a smaller household and getting more physical activity were significantly directly associated with overall HRQOL and PCS. The result also stated that the participants who had close family members diagnosed with chronic disease were significantly negatively associated with MCS only. In contrast with overall HRQOL, PCS, and MCS, there were significant associations of BP and GH in relation to sex, in which females had a negative coefficient. Furthermore, the participants who possessed a better occupational status were significantly directly associated with a health domain of SF only.

## 4. Discussion

Assessment of health-related quality of life (HRQOL) of government employees is needed in order to ensure a positive outcome in the administration and the operation of the government, as well as to optimise the well-being of the worker. This study found that government employees’ mean score of all the health domains, except RE, that incorporated HRQOL were slightly lower than the general Malaysian population reported in an earlier study [29]. As compared to a previous study, which also used SF-36 to assess HRQOL, the government employees in the present study reported lower mean scores of overall HRQOL, PCS, and MCS compared to the civil servants in China [15]. China is a more developed country than Malaysia—its science and technology keep growing to increase standard of living, worker efficiency, and labour-saving. Linearly, the study using the SF-36 mentioned that the HRQOL of the public in Malaysia was less than the public in developed countries, such as the United States, Canada, and Australia [30]. Next, the nature of the work or workplace might influence the HRQOL of the workers differently. This study found that the mean score of PCS and MCS of government employees were seen to be higher than frontline railway workers in China; nurses in Greece; bank workers in India; and workers from the transportation industry, industrial plant cooling, and universities in Singapore [11,12,31,32].

In agreement with other studies, the multivariate analysis in this study showed that increasing age was significantly associated with a decreasing HRQOL [30,33,34]. The natural aging process is interconnected with the progressive decline of physical abilities and psychological health, and increased morbidity, which can limit the productive capacity of older staff to work [16,35]. It has been suggested that older employees experience more occupation-related stress that reduce satisfaction in their health state [36]. These workplace stressors include age stereotypes and bad perceptions of older employees, such as older employees being assumed to be less adept with technology, lacking creativity, being unfit for teamwork, and having lower in emotional strength [36]. In addition, senior staff members are regularly involved in intense efforts of decision making to achieve certain goals and are supposed to fulfil high expectations from superiors in regard to their job tasks; hence, these situations might increase stressful experiences that can adversely affect the quality of life.

The present study also found that monthly personal income was significantly positively associated with HRQOL, which echoed the results of previous studies that stated that the greater one’s income, the better one’s perception of the value of health [12,37,38]. Income played a substantial role in the gradient of health disparities since it has been described that those who earn more have a lower probability of experiencing illness and premature death [39]. Psychological health was also affected by income status as people with low income reported that they were more often nervous and sad compared to people with high income [40]. To some extent, those who have a better financial situation are often in a better health state because they can more easily afford medical insurance, are able to fulfil suggested health service expenses, and are likely to experience living in residence-based health benefits [40]. To sum up, government employees who received better earnings more probably meet the costs involved in material conditions for a pleasant survival, and thus it is expressed that income directly supports HRQOL.

Another significant sociodemographic characteristic associated with HRQOL was the size of household. Similar to past studies, the current study also found that as household number increased, HRQOL decreased [12,34]. Possible explanations were that having a large family contributed to more responsibility, such as raising children; this requires spending more to provide the necessities of a comfortable life with their family members. On the other hand, a significant association between sex and HRQOL was limited to the health domains of BP and GH. This finding is consistent with previous studies that also agreed that being female was associated with poor BP and GH [41]. Working females invested more energy and faced high strain in order to cope with noticeable demanding roles, namely, responsibilities within the home and managing tasks at the workplace [42]. Plus, the majority of female workers concurred that they were used to experiencing exhaustion, murky thinking, sleep deprivation, and moodiness [42]. The present study is in line with a past study that found that a higher occupational status was significantly associated with a pleasurable health domain of SF [43]. The rationale for this result might be that employees who secured a better job position tended to receive wellness employee benefits, such as a flexible working schedule, employment leave, and worksite wellness program participation. Conjointly, the employees who possessed a superior occupational status were more likely to earn the respect from other co-workers, which can help to boost self-esteem and get involved in teamwork comfortably. Hence, such conditions may ultimately assist in increasing the well-being of social relationships.

As expected, being more physically active was significantly associated with better HRQOL. This was consistent with other studies [15,44]. Physical activity has been remarkable in reducing the risks of numerous health problems. Moreover, practising physical activity regularly strengthens the physical wellness of employees and provides adequate energy to increase productivity at work [15]. Appropriate engagement in physical activity contributes to reducing bodily stress, stimulates relaxation, improves quality of sleep, boosts the immune system, and increases social participation; thus, such circumstances directly promoted satisfaction towards the healthy state [15].

The current study also reported that the presence of chronic disease and the use of long-term medication were significantly associated with a deterioration in HRQOL. The signs of these negative associations were supported by prior studies [12,45]. It has been noted that there is high number of people diagnosed with chronic disease that riddle the overall society sectors, including the labour sector [46]. In relation, a diagnosis of one or more chronic diseases dropped job performance among employees since chronic diseases restrict physical capability, worsen inner feelings, affect career development, limit social inclusion, and increase commitment to healthcare regimes. Therefore, these situations eventually affect certain joys of life, particularly physical, psychological, and social well-being. Moreover, the findings in this study are in line with past studies that mention that a family history of chronic disease is significantly associated with poor mental health [41]. Family history has been a known risk factor of developing several chronic diseases since it reflects inherited genetic susceptibility as well as shared behaviour and environment among family members [47]. Thus, such a condition might build a feeling of worry, anxiety, and uneasiness, which negatively impacts emotional health.

## 5. Limitations and Future Research

There were limitations that we identified in this study. The design of this study was cross-sectional, which can only report association rather than causality. Furthermore, the findings from this study cannot be generalised to all the employees who work in government sectors since the study considered participants from only one state in Malaysia. Hence, it is suggested for future research to propose a better study design, such as a longitudinal study together with recruitment of a large sample size, in order to provide more accurate data. It would also be interesting for upcoming studies to include other factors, such as psychosocial factors, nutritional status, sociocultural factors, and quality of work life, for a more comprehensive understanding of factors that are associated with HRQOL among the Malaysian working population in government sectors. To our knowledge, this is the first study carried out among Malaysian government employees that has explored their HRQOL and its associated factors. Previously published studies in Malaysia point to HRQOL limited to groups of people with health issues, the elderly, those of low socio-economic status, medical students, and non-prescription medicine customers. The results from the current study can be a future reference for researchers, public health professionals, and employment policymakers to take action to guarantee the well-being of government employees.

## 6. Conclusions

In summary, government employees are reported to have slightly lower scores over most of the health domains that encompass HRQOL compared to the general Malaysian population. A multivariate analysis confirmed that the factors that are significantly associated with the SF-36 profile of scores in this study were sex, age, occupational status, monthly personal income, household number, physical activity level, presence of chronic disease, long-term medication use, and family history of chronic disease. Specifically, the present study stated that old age, low personal income status, large household size, being inactive, a chronic disease diagnosis, and long-term use of medicine result in a decrease in overall HRQOL. Older employees are more exposed to stressful experiences that can reduce their HRQOL, such as natural aging process, crucial decision making, high expectations, age stereotypes, and negative perceptions. In addition, those who earn low monthly income and live in a large household are more likely to face difficulty meeting the costs involved and need to spend more to provide the necessities of a pleasant life for their family. Consequently, these situations may affect their joy of life. Being inactive in physical activity was expected to result in poor HRQOL, since low physical activity is related to health problems, deterioration in physical strength, and unstable emotional well-being. The physical and mental health, job performance, career development, social participation, and increasing commitment towards health treatment of employees with medical problems were naturally affected, which led to the employees’ poor HRQOL. This study concluded that a family history of chronic disease is associated with poor mental health. Feelings of worry may develop because family history is acknowledged as a risk factor of chronic disease. The present study also concluded that female workers are associated with severe bodily pain (BP) and poor general health (GH), whereas those who have better occupational status experience better social functioning (SF). The HRQOL of employees plays a crucial role in increasing their productivity at work and providing a satisfactory outcome in line with the organisation’s demand. Hence, organisations and policymakers need to concentrate on improving the HRQOL of employees to strengthen their physical ability, improve emotional positivity, and create favourable social interaction at the workplace. The outcomes of this study are informative for planning intervention strategies to increase HRQOL of government employees by tackling its associated factors.

## Figures and Tables

**Table 1 ijerph-18-02626-t001:** Socio-demographic factors, lifestyle factors, and medical history of the participants.

Variables	Participants (*n* = 460) Mean ± SD *n* (%)		
	Male (*n* = 193)	Female (*n* = 267)	Total (*n* = 460)
Sex	193 (42.0)	267 (58.0)	460 (100.0)
Age (years)	35.84 ± 9.23	36.67 ± 8.45	36.32 ± 8.79
18–30	71 (36.8)	70 (26.2)	141 (30.7)
31–45	85(44.0)	151 (56.6)	236 (51.3)
46–59	37 (19.2)	46 (17.2)	83 (18.0)
Marital status			
Single/divorced/widowed	65 (33.7)	96 (36.0)	161 (35.0)
Married	128 (66.3)	171 (64.0)	299 (65.0)
Ethnicity			
Malay	168 (87.0)	229 (85.8)	397 (86.3)
Non-Malay	25 (13.0)	38 (14.2)	63 (13.7)
Educational level			
Secondary	54 (28.0)	74 (27.7)	128 (27.8)
Tertiary	139 (72.0)	193 (72.3)	332 (72.2)
Occupational status			
Professional	100 (51.8)	105 (39.3)	205 (44.6)
Support worker	93 (48.2)	162 (60.7)	255 (55.4)
Monthly personal income (MYR)	3504.19 ± 1524.26	3315.88 ± 1456.56	3394.89 ± 1486.61
<3000	79 (40.9)	119 (44.6)	198 (43.0)
3000–4999	76 (39.4)	113 (42.3)	189 (41.1)
≥5000	38 (19.7)	35 (13.1)	73 (15.9)
Monthly household income (MYR)	5603.95 ± 3085.94	5869.13 ± 3267.10	5757.87 ± 3191.62
≤4849/B40	81 (42.0)	110 (41.2)	191 (41.5)
4850–10,959/M40	99 (51.3)	138 (51.7)	237 (51.5)
≥10,960/T20	13 (6.7)	19 (7.1)	32 (7.0)
Household number	3.89 ± 2.03	3.80 ± 2.07	3.84 ± 2.05
<5 persons	103 (53.4)	147 (55.1)	250 (54.3)
≥5 persons	90 (46.6)	120 (44.9)	210 (45.7)
Physical activity level (MET-minutes/week)	3096.33 ± 3559.12	2502.39 ± 2871.71	2751.59 ± 3188.10
Low	42 (21.8)	54 (20.2)	96 (20.9)
Moderate	81 (42.0)	143 (53.6)	224 (48.7)
High	70 (36.3)	70 (26.2)	140 (30.4)
Smoking status			
Number of cigarettes (unit/day)	10.68 ± 5.95	-	10.68 ± 5.95
Current smoker	68 (35.2)	-	68 (14.8)
Non-smoker	125 (64.8)	267 (100)	392 (85.2)
Chronic disease			
None	157 (81.3)	224 (83.9)	381 (82.8)
≥1 chronic disease	36 (18.7)	43 (16.1)	79 (17.2)
Types of chronic disease			
Diabetes mellitus	11 (30.6)	10 (23.3)	21 (26.6)
Hypertension	32 (88.9)	35 (81.4)	67 (84.8)
Cardiovascular disease	-	1 (2.3)	1 (1.3)
Asthma	4 (11.1)	5 (11.6)	9 (11.4)
Hypercholesterolemia	2 (5.6)	2 (4.7)	4 (5.1)
Long-term medication use			
No	159 (82.4)	226 (84.6)	385 (83.7)
Yes	34 (17.6)	41 (15.4)	75 (16.3)
Family history of chronic disease			
No	69 (35.8)	112 (41.9)	181 (39.3)
Yes	124 (64.2)	155 (58.1)	279 (60.7)
Types of chronic diseases (family history)			
Diabetes mellitus	73 (58.9)	85 (54.8)	158 (56.6)
Hypertension	89 (71.8)	119 (76.8)	208 (74.6)
Cardiovascular disease	36 (29.0)	37 (23.9)	73 (26.2)
Chronic kidney disease	18 (14.5)	13 (8.4)	31 (11.1)
Asthma	9 (7.3)	7 (4.5)	16 (5.7)
Stroke	6 (4.8)	7 (4.5)	13 (4.7)
Hypercholesterolemia	7 (5.6)	16 (10.3)	23 (8.2)

**Table 2 ijerph-18-02626-t002:** Differences of SF-36 profile of scores by participants’ socio-demographic characteristics.

Variables	Participants (*n* = 460) Mean ± SD											
	Score of SF-36											
	PF	RP	BP	GH	VT	SF	RE	MH	HC	PCS	MCS	HRQOL
Total participants	79.04 ± 22.77	80.82 ± 27.37	64.00 ± 18.83	59.79 ± 14.91	64.92 ± 17.00	65.89 ± 16.45	86.96 ± 24.18	70.56 ± 14.13	61.52 ± 21.14	73.36 ± 17.28	71.79 ± 14.13	72.42 ± 14.99
Sex	0.549	0.673	0.053	0.044 *	0.268	0.105	0.396	0.797	0.573	0.387	0.292	0.322
Male	79.79 ± 22.48	80.18 ± 27.23	65.99 ± 19.79	61.45 ± 15.11	65.96 ± 17.36	67.36 ± 16.68	88.08 ± 23.85	70.76 ± 14.09	62.18 ± 21.97	74.19 ± 17.74	72.61 ± 14.11	73.24 ± 15.27
Female	78.50 ± 23.01	81.27 ± 27.51	62.56 ± 17.99	58.59 ± 14.68	64.18 ± 16.74	64.84 ± 16.23	86.14 ± 24.43	70.41 ± 14.18	61.05 ± 20.56	72.77 ± 16.96	71.21 ± 14.14	71.84 ± 14.79
Age (years)	<0.001 *	<0.001 *	<0.001 *	<0.001 *	<0.001 *	<0.001 *	0.002 *	<0.001 *	<0.001 *	<0.001 *	<0.001 *	<0.001 *
18–30	87.13 ± 20.82	88.48 ± 24.19	68.29 ± 18.34	64.33 ± 13.98	71.23 ± 15.43	69.77 ± 16.51	90.78 ± 23.94	75.12 ± 13.55	68.62 ± 22.64	80.16 ± 14.09	76.61 ± 12.90	78.46 ± 12.29
31–45	79.32 ± 21.58	80.72 ± 28.42	64.49 ± 17.91	60.06 ± 13.93	65.29 ± 15.14	66.31 ± 16.10	87.43 ± 24.35	70.83 ± 13.22	60.81 ± 18.87	73.59 ± 16.30	72.16 ± 13.06	72.68 ± 13.96
46–59	64.52 ± 22.41	68.07 ± 24.79	55.30 ± 19.52	51.33 ± 15.74	53.07 ± 18.54	58.13 ± 14.79	79.12 ± 22.52	62.02 ± 13.92	51.51 ± 20.42	61.18 ± 18.48	62.57 ± 14.77	61.45 ± 16.05
Marital status	<0.001 *	<0.001 *	0.002 *	0.016 *	0.001 *	0.020 *	0.178	0.008 *	<0.001 *	<0.001 *	0.003 *	<0.001 *
Single/divorced/widowed	85.19 ± 21.79	87.42 ± 24.23	67.76 ± 19.21	62.08 ± 15.55	68.39 ± 17.16	68.32 ± 16.66	89.02 ± 24.66	72.94 ± 14.67	67.08 ± 21.73	78.45 ± 15.38	74.43 ± 14.18	76.57 ± 13.79
Married	75.74 ± 22.64	77.26 ± 28.33	61.97 ± 18.33	58.56 ± 14.44	63.06 ± 16.66	64.59 ± 16.22	85.84 ± 23.89	69.27 ± 13.68	58.53 ± 20.23	70.63 ± 17.65	70.38 ± 13.92	70.19 ± 15.16
Ethnicity	0.258	0.591	0.241	0.870	0.176	0.263	0.495	0.821	0.055	0.304	0.320	0.258
Malay	78.56 ± 22.96	80.54 ± 27.71	63.59 ± 18.87	59.75 ± 14.96	64.49 ± 16.62	65.55 ± 16.27	86.65 ± 24.69	70.49 ± 13.75	60.77 ± 20.77	73.03 ± 17.39	71.53 ± 13.80	72.11 ± 14.90
Non-Malay	82.06 ± 21.47	82.54 ± 25.25	66.59 ± 18.49	60.08 ± 14.77	67.62 ± 19.22	68.06 ± 17.50	88.89 ± 20.74	70.98 ± 16.46	66.27 ± 22.97	75.45 ± 16.53	73.44 ± 16.06	74.41 ± 15.54
Educational level	0.006 *	0.014 *	0.075	0.119	0.004 *	0.072	0.064	0.025 *	0.326	0.004 *	0.005 *	0.003 *
Secondary	74.06 ± 24.84	75.78 ± 27.69	61.48 ± 18.87	58.05 ± 15.29	61.21 ± 18.32	63.67 ± 16.99	83.59 ± 25.09	68.19 ± 14.39	59.96 ± 21.36	69.38 ± 18.68	68.85 ± 15.07	68.91 ± 16.29
Tertiary	80.96 ± 21.66	82.76 ± 27.03	64.97 ± 18.75	60.47 ± 14.73	66.36 ± 16.28	66.75 ± 16.18	88.25 ± 23.73	71.47 ± 13.94	62.12 ± 21.06	74.90 ± 16.49	72.93 ± 13.60	73.78 ± 14.26

* *p*-value < 0.05 represents significance. PF: physical functioning; RP: role of physical health; BP: bodily pain; GH: general health; VT: vitality; SF: social functioning; RE: role of emotion-related health; MH: mental health; HC: health change; PCS: physical component summary; MCS: mental component summary; HRQOL: health-related quality of life.

**Table 3 ijerph-18-02626-t003:** Differences of SF-36 profile of scores by participants’ socio-economic status.

Variables	Participants (*n* = 460) Mean ± SD											
	Score of SF-36											
	PF	RP	BP	GH	VT	SF	RE	MH	HC	PCS	MCS	HRQOL
Occupational status	0.010 *	0.053 *	0.028 *	0.144	0.025 *	0.026 *	0.108	0.289	0.134	0.007 *	0.036 *	0.008 *
Professional	82.05 ± 21.29	83.54 ± 25.73	66.15 ± 18.91	60.93 ± 14.96	66.90 ± 17.62	67.80 ± 16.74	88.94 ± 21.82	71.34 ± 14.50	63.17 ± 21.66	75.79 ± 16.62	73.34 ± 14.12	74.49 ± 14.62
Support worker	76.63 ± 23.66	78.63 ± 28.48	62.27 ± 18.62	58.88 ± 14.84	63.33 ± 16.36	64.36 ± 16.08	85.34 ± 25.85	69.93 ± 13.82	60.19 ± 20.67	71.42 ± 17.59	70.56 ± 14.04	70.77 ± 15.11
Monthly personal income (MYR)	0.048 *	0.587	0.543	0.484	0.022 *	0.276	0.506	0.016 *	0.518	0.100	0.033 *	0.055
<3000	82.05 ± 22.38	81.94 ± 29.25	65.01 ± 18.61	60.76 ± 13.91	67.39 ± 15.24	67.23 ± 16.72	88.38 ± 25.68	72.71 ± 13.07	61.36 ± 21.16	75.34 ± 16.58	73.77 ± 13.18	74.34 ± 14.12
3000–4999	76.93 ± 23.23	80.69 ± 25.48	62.89 ± 18.47	59.10 ± 15.57	63.36 ± 18.26	64.55 ± 15.84	86.24 ± 24.29	68.72 ± 15.07	62.57 ± 21.20	72.06 ± 17.54	70.35 ± 14.97	71.13 ± 15.44
≥5000	76.37 ± 21.95	78.08 ± 26.99	64.11 ± 20.37	58.97 ± 15.85	62.26 ± 17.54	65.75 ± 17.18	84.93 ± 19.28	69.48 ± 13.77	59.25 ± 21.04	71.39 ± 18.17	70.19 ± 13.88	70.59 ± 15.72
Monthly household income (MYR)	0.010 *	0.035 *	0.471	0.106	0.037 *	0.098	0.422	0.024 *	0.104	0.006 *	0.033 *	0.006 *
≤4849/B40	82.77 ± 23.48	84.69 ± 27.70	65.27 ± 18.58	61.09 ± 15.18	67.17 ± 16.28	67.60 ± 16.77	88.66 ± 26.59	72.52 ± 14.64	64.01 ± 21.99	76.31 ± 16.55	73.75 ± 13.99	74.97 ± 14.11
4850–10,959/M40	76.05 ± 21.96	77.85 ± 26.64	63.14 ± 19.12	58.39 ± 14.92	62.97 ± 17.69	64.29 ± 16.05	85.94 ± 22.11	68.83 ± 13.40	59.70 ± 20.34	70.96 ± 17.55	70.17 ± 14.06	70.34 ± 15.39
≥10,960/T20	78.91 ± 21.77	79.69 ± 28.71	62.73 ± 18.21	62.34 ± 12.44	65.94 ± 14.62	67.58 ± 16.76	84.38 ± 23.92	71.63 ± 15.12	60.16 ± 20.93	73.57 ± 17.37	72.15 ± 14.35	72.65 ± 15.19
Household number	<0.001 *	<0.001 *	0.002 *	<0.001 *	<0.001 *	0.054	0.021 *	<0.001 *	<0.001 *	<0.001 *	<0.001 *	<0.001 *
<5 persons	83.06 ± 22.33	85.00 ± 26.98	66.47 ± 18.19	62.02 ± 14.49	68.36 ± 16.11	67.25 ± 16.86	89.33 ± 25.21	73.15 ± 14.58	65.00 ± 21.49	76.84 ± 16.11	74.41 ± 13.95	75.57 ± 13.97
≥5 persons	74.26 ± 22.42	75.83 ± 27.05	61.06 ± 19.19	57.14 ± 15.02	60.83 ± 17.18	64.29 ± 15.84	84.13 ± 22.64	67.47 ± 12.95	57.38 ± 19.99	69.23 ± 17.75	68.69 ± 13.73	68.69 ± 15.35

* *p*-value < 0.05 represents significance. PF: physical functioning; RP: role of physical health; BP: bodily pain; GH: general health; VT: vitality; SF: social functioning; RE: role of emotion-related health; MH: mental health; HC: health change; PCS: physical component summary; MCS: mental component summary; HRQOL: health-related quality of life.

**Table 4 ijerph-18-02626-t004:** Differences of SF-36 profile of scores by participants’ lifestyle factors and medical history.

Variables	Participants (*n* = 460) Mean ± SD											
	Score of SF-36											
	PF	RP	BP	GH	VT	SF	RE	MH	HC	PCS	MCS	HRQOL
Physical activity level (MET-minutes/week)	<0.001 *	0.002 *	0.002 *	<0.001 *	<0.001 *	0.001 *	0.016 *	0.001 *	<0.001 *	<0.001 *	<0.001 *	<0.001 *
Low	67.60 ± 24.24	72.39 ± 27.74	58.85 ± 19.67	55.05 ± 14.93	58.07 ± 17.61	61.72 ± 17.28	80.90 ± 25.93	66.58 ± 13.44	52.86 ± 20.48	64.69 ± 18.59	66.53 ± 14.29	65.08 ± 15.91
Moderate	79.78 ± 21.24	82.14 ± 25.92	64.03 ± 18.32	59.08 ± 14.49	65.11 ± 16.62	65.18 ± 15.74	87.79 ± 22.54	70.30 ± 13.87	61.38 ± 20.34	73.80 ± 16.36	71.84 ± 13.79	72.69 ± 14.45
High	85.71 ± 21.23	84.46 ± 28.36	67.48 ± 18.37	64.18 ± 14.48	69.32 ± 15.76	69.91 ± 16.21	89.76 ± 24.94	73.69 ± 14.37	67.68 ± 20.88	78.61 ± 15.53	75.34 ± 13.49	77.04 ± 13.25
Smoking status	0.003 *	0.389	0.087	0.005 *	0.053	0.121	0.801	0.403	0.023 *	0.007 *	0.239	0.020 *
Current smoker	85.88 ± 19.92	83.46 ± 25.78	67.61 ± 19.44	64.49 ± 14.48	68.60 ± 16.23	68.75 ± 15.19	86.27 ± 28.35	71.88 ± 13.84	66.91 ± 20.91	78.59 ± 15.68	73.58 ± 13.88	76.32 ± 13.99
Non-smoker	77.86 ± 23.05	80.36 ± 27.64	63.37 ± 18.67	58.98 ± 14.86	64.29 ± 17.08	65.40 ± 16.63	87.07 ± 23.42	70.33 ± 14.19	60.59 ± 21.07	72.46 ± 17.41	71.49 ± 14.16	71.75 ± 15.08
Chronic disease	<0.001 *	<0.001 *	<0.001 *	<0.001 *	<0.001 *	<0.001 *	<0.001 *	<0.001 *	<0.001 *	<0.001 *	<0.001 *	<0.001 *
None	83.56 ± 20.80	85.04 ± 26.02	66.91 ± 18.05	62.66 ± 13.75	68.16 ± 15.24	68.34 ± 15.87	90.20 ± 22.49	73.18 ± 12.97	64.44 ± 20.64	77.28 ± 14.93	74.70 ± 12.42	75.92 ± 12.78
≥1 chronic disease	57.28 ± 19.08	60.44 ± 24.55	49.97 ± 16.03	45.95 ± 12.38	49.30 ± 16.52	54.11 ± 13.97	71.31 ± 26.00	57.92 ± 12.65	47.47 ± 17.72	54.49 ± 15.37	57.78 ± 13.51	55.57 ± 13.44
Long-term medication use	<0.001 *	<0.001 *	<0.001 *	<0.001 *	<0.001 *	<0.001 *	<0.001 *	<0.001 *	<0.001 *	<0.001 *	<0.001 *	<0.001 *
No	82.71 ± 21.31	84.42 ± 26.65	66.20 ± 18.16	61.94 ± 14.09	67.86 ± 15.33	67.66 ± 16.10	89.61 ± 23.48	72.89 ± 13.41	63.77 ± 20.62	76.52 ± 15.57	74.29 ± 12.93	75.29 ± 13.31
Yes	60.20 ± 20.75	62.33 ± 23.38	52.70 ± 18.24	48.80 ± 14.21	49.87 ± 17.34	56.83 ± 15.28	73.33 ± 23.25	58.56 ± 11.44	50.00 ± 20.13	57.18 ± 16.67	58.99 ± 13.14	57.69 ± 14.56
Family history of chronic disease	0.563	0.100	0.340	0.008 *	0.002 *	0.252	0.069	0.003 *	0.393	0.133	0.003 *	0.032 *
No	79.81 ± 22.99	83.43 ± 28.16	65.04 ± 17.58	62.07 ± 13.76	67.79 ± 14.95	66.99 ± 16.07	89.50 ± 23.97	72.88 ± 12.59	62.57 ± 21.35	74.87 ± 16.52	74.15 ± 12.56	74.25 ± 13.88
Yes	78.55 ± 22.66	79.12 ± 26.76	63.32 ± 19.59	58.32 ± 15.46	63.06 ± 18.00	65.19 ± 16.68	85.30 ± 24.22	69.05 ± 14.87	60.84 ± 21.02	72.39 ± 17.72	70.27 ± 14.88	71.25 ± 15.58

* *p*-value < 0.05 represents significance. PF: physical functioning; RP: role of physical health; BP: bodily pain; GH: general health; VT: vitality; SF: social functioning; RE: role of emotion-related health; MH: mental health; HC: health change; PCS: physical component summary; MCS: mental component summary; HRQOL: health-related quality of life.

**Table 5 ijerph-18-02626-t005:** Associations between socio-demographic factors, lifestyle factors, and medical history with health-related quality of life of the participants.

Variables	Participants (*n* = 460) Standardised Regression Coefficients											
	PF	RP	BP	GH	VT	SF	RE	MH	HC	PCS	MCS	HRQOL
Sex												
Male (ref.)	-	-	ref.	ref.	-	ref.	-	-	-	-	-	-
Female	-	-	−0.102	−0.099	-	−0.068	-	-	-	-	-	-
*p*-value	-	-	0.021 *	0.019 *	-	0.125	-	-	-	-	-	-
Age (years)	−0.165	−0.150	−0.096	−0.076	−0.242	−0.147	−0.039	−0.213	−0.090	−0.189	−0.215	−0.201
*p*-value	0.004 *	0.003 *	0.057	0.120	<0.001 *	0.004 *	0.444	<0.001 *	0.113	<0.001 *	<0.001 *	<0.001 *
Marital status												
Single/divorced/widowed (ref.)	ref.	ref.	ref.	ref.	ref.	ref.	ref.	ref.	ref.	ref.	ref.	ref.
Married	−0.050	−0.071	−0.084	−0.023	−0.017	0.010	−0.004	−0.004	−0.063	−0.057	−0.008	−0.033
*p*-value	0.337	0.148	0.061	0.585	0.716	0.832	0.936	0.925	0.245	0.246	0.856	0.498
Ethnicity												
Malay (ref.)	-	-	ref.	-	ref.	-	-	-	ref.	-	-	-
Non-Malay	-	-	0.081	-	0.080	-	-	-	0.091	-	-	-
*p*-value	-	-	0.064	-	0.054	-	-	-	0.060	-	-	-
Educational level												
Secondary (ref.)	ref.	ref.	ref.	ref.	ref.	ref.	ref.	ref.	-	ref.	ref.	ref.
Tertiary	0.024	0.066	0.048	0.021	0.033	−0.012	0.057	−0.005	-	0.004	0.021	0.008
*p*-value	0.593	0.134	0.271	0.622	0.475	0.824	0.207	0.909	-	0.918	0.644	0.852
Occupational status												
Professional (ref.)	ref.	ref.	ref.	ref.	ref.	ref.	ref.	-	ref.	ref.	ref.	ref.
Support worker	−0.044	−0.071	−0.070	−0.025	−0.045	−0.086	−0.056	-	−0.037	−0.022	−0.026	−0.022
*p*-value	0.366	0.106	0.111	0.549	0.350	0.049 *	0.210	-	0.398	0.632	0.589	0.634
Monthly personal income (MYR)	0.134	-	-	-	0.118	-	-	0.129	-	0.169	0.128	0.173
*p*-value	0.005 *	-	-	-	0.012 *	-	-	0.007 *	-	<0.001 *	0.006 *	<0.001 *
Monthly household income (MYR)	0.018	0.057	-	0.042	0.009	0.043	-	0.025	0.102	0.005	0.004	0.024
*p*-value	0.819	0.254	-	0.323	0.902	0.420	-	0.719	0.057	0.949	0.952	0.740
Household number (persons)	−0.096	−0.057	−0.078	−0.082	−0.066	0.028	−0.017	−0.052	−0.138	−0.112	−0.045	−0.095
*p*-value	0.046 *	0.258	0.082	0.058	0.170	0.573	0.719	0.286	0.002 *	0.014 *	0.349	0.034 *
Physical activity level (MET-minutes/week)	0.119	0.002	0.079	0.100	0.056	0.054	0.041	0.047	0.111	0.100	0.061	0.095
*p*-value	0.005 *	0.970	0.076	0.020 *	0.188	0.232	0.366	0.280	0.013 *	0.013 *	0.150	0.018 *
Smoking status												
Current smoker (ref.)	ref.	-	ref.	ref.	ref.	ref.	-	-	ref.	ref.	ref.	ref.
Non-smoker	−0.064	-	0.002	−0.042	−0.034	−0.028	-	-	−0.066	−0.049	0.004	−0.041
*p*-value	0.120	-	0.968	0.392	0.410	0.522	-	-	0.133	0.221	0.932	0.292
Chronic disease												
None (ref.)	ref.	ref.	ref.	ref.	ref.	ref.	ref.	ref.	ref.	ref.	ref.	ref.
≥1 chronic disease	−0.253	−0.265	−0.343	−0.408	−0.205	−0.249	−0.295	−0.216	−0.255	−0.338	−0.265	−0.305
*p*-value	<0.001 *	<0.001 *	<0.001 *	<0.001 *	<0.001 *	<0.001 *	<0.001 *	<0.001 *	<0.001 *	<0.001 *	<0.001 *	<0.001 *
Long-term medication use												
No (ref.)	ref.	ref.	ref.	ref.	ref.	ref.	ref.	ref.	ref.	ref.	ref.	ref.
Yes	−0.122	−0.111	−0.075	−0.090	−0.176	−0.033	−0.097	−0.167	−0.072	−0.126	−0.159	−0.151
*p*-value	0.026 *	0.057	0.195	0.106	0.001	0.569	0.102	0.003 *	0.218	0.015 *	0.004 *	0.003 *
Family history of chronic disease												
No (ref.)	-	ref.	-	ref.	ref.	-	ref.	ref.	-		ref.	ref.
Yes	-	−0.043	-	−0.011	−0.006	-	−0.014	−0.007	-	−0.008	−0.096	−0.069
*p*-value	-	0.355	-	0.806	0.893	-	0.761	0.867	-	0.853	0.019 *	0.088
R^2^ value	0.243	0.128	0.122	0.195	0.227	0.124	0.085	0.118	0.130	0.246	0.320	0.329

* *p*-value < 0.05 represents significance. Ref.: reference group. PF: physical functioning; RP: role of physical health; BP: bodily pain; GH: general health; VT: vitality; SF: social functioning; RE: role of emotion-related health; MH: mental health; HC: health change; PCS: physical component summary; MCS: mental component summary; HRQOL: Health-related quality of life.

## Data Availability

The data presented in this study are available on request from the corresponding author.

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
