# Peer review of "Factors Associated with Health-Related Quality of Life among Government Employees in Putrajaya, Malaysia"

_ijerph, 2021, doi:10.3390/ijerph18052626_

Round 1

Reviewer 1 Report

In its current form, the manuscript is still in a very initial stage, pretty far from the quality standards of an academic paper. So, I recommend a minor revision. I explain some of my reservations in detail below

  1. Introduction: The introduction should be much more focused. The research objectives should be much clearer. Perhaps it could be helpful to explicitly articulate the research question. Similarly, authors need to clearly state the value-added of the paper and better discuss how this work could be worth it for both academics and practitioners. I suggest you explain the answer to below mentioned four questions in your revised introduction part. Below mentioned studies will help you to re-write your introduction. I suggest fallow these studies and cite in your manuscript.
  2. What are the academic research questions of this study?
  3. I suggest to the authors in the last paragraph of the introduction explain the structure of the paper.
  • I also suggest you write your research background as a story
  1. Zhou, X.; Rasool, S.F.; Ma, D. The Relationship between Workplace Violence and Innovative Work Behavior: The Mediating Roles of Employee Wellbeing. Healthcare 2020, 8, 332.
  2. Rasool, S.F.; Wang, M.; Zhang, Y.; Samma, M. Sustainable Work Performance: The Roles of Workplace Violence and Occupational Stress. International journal of environmental research and public health 2020, 17, 912.
  3. Literature Review: Literature Review of this study is missing from the paper. After the introduction, you have written a direct material and methods. I suggest you made a new heading with the title of “Literature Review.” Under this heading, explain the literature review of your study. Below mentioned studies will help you to write your literature review part.

Rasool, S.F.; Maqbool, R.; Samma, M.; Zhao., Y.; Anjum, A. (2019). Positioning Depression as a Critical Factor in Creating a Toxic Workplace Environment for Diminishing Worker Productivity. Sustainability, 11 (9), 2589 (1-18).

  1. Materials and Methods: The validation of the research is doubtful. Please, provide more details about the research instrument. Moreover, provide a detailed questionnaire as an appendix at the end of the paper.
  2. Results: This section is well explained
  3. Discussion: The discussion section is also well explained.
  4. Conclusion: This section is feeble; it is suggested to integrate the conclusion with the discussion and literature of the study.
  5. Limitations and future research: The limitations of this study are missing I suggest you develop the last heading with the title of the “limitations and future research” and explained it in detail.
  6. References: It is recommended to make use of recent references to support these arguments (ideally, published during the past 5 years).

Reviewer 2 Report

The paper is generally well written. One advantage is using the standard SF-36 survey instrument and a well-defined original population of governmental employees. I only suggest some clarifications and better editing of the tables. The tables are unreadable due to an overabundance of information. By default, only the main PCS, MCS and HRQOL indices can be presented using the SF-36. I propose to move the data on detailed domains to the electronic appendix. This is all the more justified as the authors do not describe these detailed results, except for the gender results.  

Procedure: The sentence about checking questionnaires after completion is unclear (line 96-97). Who did the checking? Does this not interfere with anonymity. 

Analysis: In the regression analysis, categorized variables were recoded to 0-1, which is associated with loss of information. Alternatively, generalized linear models can be used, or recoding into two dummies (classical regression). Please address the methodological simplifications in the discussion in the limitations section. Please describe in the methods which variables remained continuous (probably age, income, no of household members and PA) and indicate that the rest are dichotomous. Linear regression models should definitely have R-square given. At least for mean indices suggested to be left in main text. The sentences on lines 165-167 are unclear and contradictory to the further presentation of the results. Non-significant variables appear also in the final models. 

Tables: As mentioned, the tables are not very readable and too long. In the basic text, the three main SF-36 indices would have been adequate; in the current version they get lost at the end of the table.  Tables 2 could be divided into three, corresponding to different groups of variables (sociodemographic, wealth and employment, health status). Reducing the number of columns (HRQL subdimensions) will allow the p-value to be placed in the next column to results (tab.2 and tab3.), further reducing the length of the tables. Currently, in Table 2, the p-values are in the rows above the results without explanation and it is difficult to guess what is meant. The explanation of the abbreviation of domains names should be at least under the first table. However, better keep detailed domains in annex.   

The list of references is long enough but too local, dominated by papers from Malaysia. Neither in the introduction nor in the discussion is it clearly highlighted that this type of research is also being done in other countries.  The method of citation is too mechanical. It looks like a citation based on the title of the paper, without reference to its content, differences and similarities, relative to the results obtained.

In the notation of references, the names of the journals should be checked and the inconsistent use of capital letters in the names should be corrected.  

Reviewer 3 Report

Thank you for doing this important work. Is a current topic of great importance. I hope that the following points will help you to strengthen your work before publication.

Title

I recommend removing the abbreviation "HRQOL" from the title

Abstract

Avoid the sentence “The results of this study signify that the mean score of overall HRQOL was 72.42 ± 14.99” Readers will not understand the mean values ​​before reading the text. Reframe this result

Materials and Methods

Could you indicate what type of public employees the authors studied? What area?

What was the response rate?

How long did the questionnaire take?

Who collected, delivered and collected the questionnaires?

How was the informed consent given?

Reviewer 4 Report

Congratulate the authors for the work done, but it is necessary to clarify certain sections. - The introduction should better contextualize the labor characteristics of government employees, number of working hours, shifts, remuneration as well as eventuality in the categories studied. - The method should show the total number of government employees, how the samples were clearly selected, response rate and the code of the ethics committee. - Tables must be rewritten if you lose one in their reading and analysis, subdivide them in the order that exposes the results in the text. Put a legend with the abbreviations you use in them. You don't know what you are analyzing a table because it lacks an explanation in the columns, rewrite it in full (table 2). - Once the results have been rewritten, the discussion should follow that order, providing recent studies that discuss each section of the results. - Bibliographic References, 18/49 citations are older than 5 years, and should be replaced by more recent citations, which after a search in pubmed this reviewer has found more recent data that would increase the quality of his work.

Round 2

Reviewer 4 Report

The changes requested by this reviewer have been made. the quality and presentation have improved considerably with the modifications.